# The Emerging Evidence for a Protective Role of Fucoidan from *Laminaria japonica* in Chronic Kidney Disease-Triggered Cognitive Dysfunction

**DOI:** 10.3390/md20040258

**Published:** 2022-04-07

**Authors:** Zhihui Ma, Zhiyou Yang, Xinyue Feng, Jiahang Deng, Chuantong He, Rui Li, Yuntao Zhao, Yuewei Ge, Yongping Zhang, Cai Song, Saiyi Zhong

**Affiliations:** 1Guangdong Provincial Key Laboratory of Aquatic Product Processing and Safety, Guangdong Province Engineering Laboratory for Marine Biological Products, Guangdong Provincial Engineering Technology Research Center of Seafood, Key Laboratory of Advanced Processing of Aquatic Product of Guangdong Higher Education Institution, College of Food Science and Technology, Guangdong Ocean University, Zhanjiang 524088, China; mazhihui@stu.gdou.edu.cn (Z.M.); 2112003030@stu.gdou.edu.cn (X.F.); 2112003070@stu.gdou.edu.cn (J.D.); 2112103046@stu.gdou.edu.cn (C.H.); lirui@gdou.edu.cn (R.L.); zhaoyt@gdou.edu.cn (Y.Z.); zhangyp@gdou.edu.cn (Y.Z.); cai.song@dal.ca (C.S.); zhongsy@gdou.edu.cn (S.Z.); 2Collaborative Innovation Center of Seafood Deep Processing, Dalian Polytechnic University, Dalian 116034, China; 3Key Laboratory of Digital Quality Evaluation of Chinese Materia Medica of State Administration of TCM, Guangdong Pharmaceutical University, Guangzhou 510006, China; geyuewei@gdpu.edu.cn

**Keywords:** chronic kidney disease, fucoidan, cognitive dysfunction, neuroinflammation, oxidative stress, GSK3β-Nrf2-HO-1 signaling, microglial polarization

## Abstract

This study aimed to explore the mechanism of fucoidan in chronic kidney disease (CKD)-triggered cognitive dysfunction. The adenine-induced ICR strain CKD mice model was applied, and RNA-Seq was performed for differential gene analysis between aged-CKD and normal mice. As a result, fucoidan (100 and 200 mg kg^−1^) significantly reversed adenine-induced high expression of urea, uric acid in urine, and creatinine in serum, as well as the novel object recognition memory and spatial memory deficits. RNA sequencing analysis indicated that oxidative and inflammatory signaling were involved in adenine-induced kidney injury and cognitive dysfunction; furthermore, fucoidan inhibited oxidative stress via GSK3β-Nrf2-HO-1 signaling and ameliorated inflammatory response through regulation of microglia/macrophage polarization in the kidney and hippocampus of CKD mice. Additionally, we clarified six hallmarks in the hippocampus and four in the kidney, which were correlated with CKD-triggered cognitive dysfunction. This study provides a theoretical basis for the application of fucoidan in the treatment of CKD-triggered memory deficits.

## 1. Introduction

Chronic kidney disease (CKD) is a chronic renal structural and functional deficit caused by multiple factors. Over the past 10 years, CKD blossomed into an important cause of morbidity and mortality worldwide with a cumulative mortality rate of 20.6% [1]. Currently, approximately 8–16% of the global adult population suffer from CKD, while the prevalence rate of CKD increases continuously along with aging and mainly by the presence of other risk factors, such as diabetes mellitus, obesity, metabolic syndrome, arterial hypertension, and indiscriminate use of nephrotoxic drugs [2]. Recently, cognitive impairment in patients, accompanied with CKD progressing, has attracted more and more attention. Cognitive dysfunction occurs not only in patients with end-stage renal disease, but also in early stage I–II CKD patients [3]. An epidemiological study demonstrated that kidney dysfunction and albuminuria are independent risk factors for the development of dementia [4]. Reports evidenced that the incidence of cognitive impairment in children and adults with CKD is higher than that in normal people, and the subjects with kidney dysfunction are always accompanied with less cortical thickness and smaller hippocampal volume and gray matter volume than those with normal renal function [5,6]. Once cognitive impairment occurs, the resulting low medication compliance of CKD patients affects their clinical treatment efficacy and increases the risk of death [7]. However, the molecular mechanism of CKD-induced cognitive deficits remains to be elucidated, and additionally, medications are urgently needed to improve CKD-induced cognitive impairment.

A variety of bioactive molecules, including polyunsaturated fatty acids (PUFA), polysaccharides, minerals, vitamins, antioxidants, enzymes, and peptides, are derived from marine organisms [8]. Marine algae, as a new source of bioactive substances, has attracted extensive attention [9]. Currently, worldwide different kinds of algae have been focused on and developed into functional diet and health products [10]. Fucoidan, sulfated polysaccharides from brown algae, are rich in fucose and sulfate groups, along with small proportions of mannose, glucuronic acid, glucose, xylose, arabinose, and rhamnose. The fucoidan extracted from *Fucus vesiculosus* contains a backbone of alternating (1 → 3)-linked α-l-fucopyranose and (1 → 4)-linked α-l-fucopyranose residues, the presence of sulfate groups mainly at *O*-2 and at a lesser extent at *O*-3 [11]. Reports have shown that fucoidan exerts multiple biological functions, such as neuroprotection, anti-tumor, anti-inflammatory, anti-virus, antioxidation, anticoagulant, hypolipidemic, and antithrombotic activity, as well as effects against hepatic, renal, and uropathic disorders [12,13]. Fucoidan ameliorated adenine-induced renal dysfunction and renal tubulointerstitial damage through regulating the FGF23–Klotho signaling axis and ERK1/2-SGK1-NHERF-1-NaPi-2a pathway in chronic kidney disease [14]. In addition, the low molecular weight fucoidan protected kidneys from dysfunction and fibrogenesis by inhibiting the TGF-β pathway in a diabetic nephropathy (DN) rat model [15]. Nevertheless, whether fucoidan ameliorated memory deficits in adenine-induced CKD remains to be elucidated.

In the current study, we investigated the differential genes between aging-related CKD and normal young mice using RNA-Seq analysis, and oxidative and inflammatory signaling pathways involved in CKD were clarified based on GO analysis. Furthermore, the effects of fucoidan on adenine-induced memory deficits and possible signaling pathways were determined. The experimental results indicated that fucoidan could ameliorate memory deficits in CKD, which indicates that fucoidan is a likely candidate drug for CKD intervention.

## 2. Results

### 2.1. Fucoidan Ameliorated Renal Function in Adenine-Induced CKD Mice

We previously investigated the effects of fucoidan on SH-SY5Y neuronal cells, and the dosage less than 1 mg mL^−1^ did not significantly affect the neuronal viability [12]. We orally co-administered fucoidan (Fuc) and adenine to mice (male, 8-week-old) for 33 days; the selected doses referred to previous published research [16]. The mice were immediately sacrificed after the behavioral test, and kidney, serum, and urine were collected. On day 33, the mice body weight, kidney index (ratio of kidney to body weight), and weight variable rate (the percentage of body changed within Fuc treatment) in the vehicle group were significantly decreased compared with control group (Figure 1A). The body weight and kidney index were not reversed by the low (Fuc-L, 100 mg kg^−1^ d^−1^) and high (Fuc-H, 200 mg kg^−1^ d^−1^) dose of Fuc treatment, or by the the losartan (Los, positive control, 20 mg kg^−1^ d^−1^) (Figure 1A). The weight variable rate showed an increased trend after Fuc-H treatment (Figure 1B).

In the locomotion test, adenine treatment dramatically decreased the moved distance compared with control mice, while the drug-treated groups failed to reverse it (Figure 1C). No significant locomotor differences were detected in the distances moved between fucoidan- and vehicle-treated groups. The mice body weights were recorded during the drug administration, with no significant changes observed in fucoidan-treated groups compared with the vehicle group (Appendix A). The results indicated that fucoidan treatment did not significantly affect the general health and behavior of the mice. The high expressed levels of uric acid and urea are important indicators for evaluation of renal failure [17]. Creatinine, a nitrogenous end product of protein metabolism, is a valuable marker for renal function [18]. CKD triggered profound histological changes, including glomeruli damages, tubular necrosis, and infiltration of inflammatory cells in kidneys, which was accompanied with a loss of renal functions indicated by the increase of urea, creatinine, and uric acid levels [19]. Thus, we determined the expressed levels of uric acid, urea, and creatinine. Compared with the control group, the uric acid and urea levels in urine were remarkably increased in adenine-treated vehicle mice, and creatinine in serum showed an increased trend, while both fucoidan (100 and 200 mg kg^−1^ d^−1^) and losartan treatment dramatically decreased the levels of uric acid, urea, and creatinine (Figure 1D–F). In addition, we investigated the relatively low dose (10 mg kg^−1^ d^−1^) of fucoidan on adenine-triggered chronic kidney disease and memory deficits and found that the renal as well as memory function were also significantly improved (data not shown).

### 2.2. Fucoidan Ameliorated Cognitive Deficits in Adenine-Induced CKD Mice

To explore the effects of fucoidan on CKD-triggered recognition memory deficits, we performed a novel object recognition test, object location test, and passive avoidance memory test, as described previously [20]. In the novel object recognition memory test, the mice equally explored the identical objects A and A’ (preferential index approximately 50%). However, in the test session, compared with vehicle mice, fucoidan and losartan treatment significantly increased the exploration times towards novel object B, which was also observed in the control group, and indicated the recognition memory was enhanced (Figure 2A).

To test whether spatial memory is improved by fucoidan, we performed the object location memory test. The preferential index was approximately 50% among groups in the training session (Figure 2B). In the test session, compared with control group, the object location memory was remarkably decreased in CKD mice, which was completely reversed after fucoidan and losartan treatment (Figure 2B). In the passive avoidance memory test, the step-through latency of mice entering the darkroom in the vehicle group was shortened and the number of errors into the darkroom was significantly increased compared with normal control mice (Figure 2C,D). After treatment with fucoidan and losartan, the step-through latency was elongated, while the number of errors were dramatically decreased (Figure 2C,D).

Collectively, these results indicated that fucoidan rescues cognitive deficits when orally administered to adenine-induced CKD mice.

### 2.3. Fucoidan Regulated Brain and Kidney Genes in Aging Renal Failure Mice

To investigate the key factors involved in CKD-triggered memory deficits, we performed RNA-Seq analysis using aging-induced CKD mice (male, 24-month-old) to mimic CKD conditions in human beings. As evidenced by ELISA analysis of uric acid and urea, the aging mice showed significant increased expression compared with young mice (Appendix A). The differential genes were determined with log2Fold change <−1 or >1, and FDR < 0.05 means significant difference. Compared with normal young mice (male, 3-month-old), 307 genes were upregulated and 189 genes were downregulated in the brain (Appendix A), while 835 genes were upregulated and 403 genes were downregulated in the kidney (Appendix A).

We performed GO annotation and enrichment analysis to explore the signaling pathways involved in the brain–kidney axis, which contributed to CKD-triggered memory deficits. The obtained differential genes between aged-CKD mice and normal young mice were used for GO enrichment analysis through the DAVID database. In GO annotation analysis, the regulated molecular functions include antioxidant activity, binding, cargo receptor activity, catalytic activity, and others, in kidneys (Figure 3A) as well as in the brain (Figure 3B). The GO enrichment analysis results indicated that 30 top biological processes are mainly involved, and among them inflammation-related signaling, such as immune system response, inflammatory response, B-cell-mediated immunity, innate immune response, and regulation of cytokine production, are widely regulated in naturally aging renal failure mice brain and kidneys (Figure 4A,B). Thus, we focused on the oxidative and inflammatory changes in CKD mice.

### 2.4. Fucoidan Ameliorated Oxidative Stress Via Nrf2-HO-1 Signaling Pathway in Adenine-Induced CKD Mice

Oxidative stress is harmful to cells due to excessive generation and accumulation of reactive oxygens (ROS), free radicals, and nitrogen. The kidney, a highly metabolic organ, is rich in oxidation reactions in mitochondria. Thus, the kidney is fragile and can be damaged by oxidative stress, and reports have shown that oxidative stress accelerates the progression of CKD [21]. Molecules targeting Nrf2-HO-1 signaling showed potential protective effects against oxidative stress and inflammation in CKD [22]. Cytokine-induced damage and oxidative stress play critical roles in the kidney–brain crosstalk during acute and chronic kidney injury [23]. MDA, the end product of lipid oxidation, was dramatically increased in CKD mice brain hippocampus and kidney compared with control mice. After fucoidan treatment as well as losartan intervention, the expression level of MDA was significantly reversed (Figure 5A,D). Activation of GSH-Px and SOD, the main antioxidant enzymes, is an important therapeutic strategy for the treatment of diseases characterized by elevated oxidative stress [24]. The activities of GSH-Px and SOD in CKD mice hippocampus and kidney were remarkably downregulated, while they were attenuated in mice hippocampus and kidney, respectively (Figure 5B,C,E,F).

We tested whether the GSK3β-Nrf2-HO-1 signaling pathway is involved in fucoidan-mediated antioxidative effects. Compared with control group, the expressions of GSK3β mRNA both in hippocampus and kidney were significantly increased in the vehicle group, which was ameliorated after fucoidan and losartan treatment (Figure 6A,E). Nuclear factor-E2 related factor 2 (Nrf2), a redox sensitive transcription factor, has been considered as a downstream target of GSK-3β [25]. Nrf2 can bind to antioxidant response elements (Are) in the nucleus and activate the expression of antioxidant genes, such as heme oxygenase-1 (HO-1) and NADPH quinone oxidoreductase 1 (NQO1) [26,27]. Compared with control group, the mRNA expressions of Nrf2 (Figure 6B,F), HO-1 (Figure 6C,G), and NQO1 (Figure 6D,H) in hippocampus and kidney were significantly decreased in the vehicle group, which were reversed by fucoidan treatment.

### 2.5. Fucoidan Inhibited Inflammatory Response in Adenine-Induced CKD Mice

We investigated whether fucoidan regulated the release of pro-inflammatory cytokines. As a result, fucoidan significantly inhibited the mRNA expression of TNF-α (Figure 7A,D) and IL-1β (Figure 7B,E) and promoted the mRNA expression of anti-inflammatory cytokine IL4 (Figure 7C,F) in the hippocampus and kidney of adenine-induced CKD mice. Reports have shown that the pro-inflammatory cytokines were regulated via microglial/macrophage polarization [28]. CD11b and inducible nitric oxide synthase (iNOS) are mainly expressed markers in M1 microglia/macrophage, while arginase 1, TGFβ, and CD206 are expressed in M2 microglia/macrophage [29]. The results convinced that fucoidan downregulated M1 microglia/macrophage-related mRNA expression of iNOS (Figure 8A,E), and on the contrary, fucoidan upregulated M2 microglia/macrophage-related mRNA expression of TGFβ (Figure 8B,F), Arg1 (Figure 8C,G), and CD206 (Figure 8D,H) in the hippocampus and kidney, respectively.

The results indicated that fucoidan exerts anti-inflammatory effects via inhibiting M1 microglial/macrophage polarization and promoting M2 microglial/macrophage polarization.

### 2.6. Fucoidan Attenuated Cognitive-Behavior-Related Hallmarks in Adenine-Induced CKD Mice

To clarify the hallmarks involved in CKD-triggered memory deficits, we reanalyzed the differential genes between aged-CKD mice and young wild mice. In the brain, Alzheimer’s disease and memory-related genes (Camk2n1, Apod, and Serpina3n), and immune-response-related genes (Spp1, CD68, GFAP, Mpeg, Ddx3y, Lyz2, CTSZ, Tyrobp, C4b, and C1qa), were selected for mRNA qPCR analysis in adenine-induced CKD mice. In the kidney, catalysis and metabolic-function-related genes (Acy3 and Angptl7), ion transporters (Slc22a12, Slco1a1, and Slc7a13), and immune-response-related genes (Cxcl13, Chd9, and Spp1) were investigated. The results were shown in Appendix A, and the Pearson correlation analysis between gene expression and novel object recognition memory was performed. We found that the Camk2n1 gene in the hippocampus exerts obvious positive correlation with cognitive behavior (Figure 9), and SPP1, Cd68, GFAP, Apod, and Serpina3 exert negative correlation (Figure 9B–F). Additionally, Acy3 and Slc22a12 in the kidney showed positive correlation with cognitive behavior (Figure 10A,B), while Angptl7 and Cxcl13 presented negative correlation (Figure 10C,D). The other genes in the hippocampus and kidney were not correlated with cognitive memory (Appendix A).

## 3. Discussion

According to glomerular filtration rate (GFR), CKD is divided into five stages, among which end-stage renal disease (ESRD) appears directly associated with the risk of dementia [23,30]. Investigations suggested that uremic neurotoxins play crucial roles in CKD-associated mild cognitive impairment, uremic neurotoxins are harmful to the brain monoaminergic system, and that this system is involved in the altered sleep pattern commonly observed in patients with CKD [31]. Uremic neurotoxins enter the brain via the blood–brain barrier and cause oxidative stress and inflammation, as well as deteriorate neuronal cell damage. In addition, uremic neurotoxins amplify the dysregulation of sodium, potassium, and water channels in the kidney, accelerating their own accumulation and blood circulation [23]. The current therapies targeting cognitive improvement in CKD mainly include dialysis treatments and neuroinflammatory and nutritional interventions. Unfortunately, the partial clearance of blood toxins with hemodialysis, as well as the limited effects of nutritional intervention, seem unlikely to prevent or slow the progression of MCI to dementia, [31]. Fucoidan is a natural sulfated polysaccharide and has been proven to exert multiple functions, especially for renal protection. We found that fucoidan attenuates adenine-induced high expression of urea, uric acid, and serum creatinine, as reported by other groups. Nevertheless, it was the first report that fucoidan ameliorates CKD-triggered recognition memory and spatial memory deficits.

In the current study, a relatively high concentration of fucoidan was adopted; thus, the toxicity has become an issue that cannot be ignored. An acute toxic study of fucoidan from *Sargassum Wightii* grevillea in Wistar rats showed that a 2000 mg kg^−1^ dose of fucoidan did not induce any remarkable toxic signs or mortality, while in the subacute toxicity study, except for a reduction of serum glucose and cholesterol, no significant changes of biochemical, hematological parameters, or toxic signs at organ levels were observed after daily oral administration of 100–400 mg kg^−1^ fucoidan for 28 days [32]. The 28-day oral administration of low molecular weight of fucoidan (<667 Da) from *Laminaria japonica* up to 2000 mg kg^−1^ caused no toxicological indications in Sprague-Dawley rats [33]. However, the 6-month treatment of fucoidan from *Laminaria japonica* at doses of 900 and 2500 mg kg^−1^ significantly prolonged the clotting time [34]; the same phenomenon was observed after gavage of fucoidan from *Cladosiphon okamuranus* over 1200 mg kg^−1^ for 3 months [35]. The aforementioned differential results indicate that the potential toxicity of fucoidan depends on the originated sources and doses, as well as the molecular weights. The safety of fucoidan was also evaluated in humans, and no changes in the liver and kidney parameters were observed after oral ingestion of 4.05 g per day of mozuku fucoidan derived from *Cladosiphon okamuranus* [36]. Our results indicated that 33-day oral administration of fucoidan (200 mg kg^−1^) did not affect the body weights and locomotor activity compared with the adenine-treated vehicle group. Nevertheless, the acute and subacute toxicity should be investigated in a future study. Losartan, an angiotensin II type 1 (AT1) receptor antagonist, was approved by the FDA in 1995 for hypertension and renal disease. Recently, losartan has been demonstrated to exert neurogenesis, astrocyte motility, and memory improvement effects in APP/PS1 AD model mice [37]. Losartan significantly improved memory function in 75–89-year-old elderly hypertensive patients [38]. Intriguingly, 30-day treatment of losartan led to a hypotension model in rats, and promoted dendritic spine loss, tau hyperphosphorylation, and memory deficits [39], indicating blood pressure as a considerable factor when taking losartan. Our study demonstrated that losartan (20 mg kg^−1^) treatment reversed adenine-induced memory deficits. Although a low dose of fucoidan (10 mg kg^−1^) in our parallel study showed memory attenuation in adenine-induced CKD mice (data not shown), a further investigation should be performed to compare the effect of fucoidan and losartan on cognitive function.

RNA sequencing analysis is an efficient way to explore possible signaling pathways. The GO annotation and enrichment analysis of significantly differential genes in naturally aging renal failure mice indicated that oxidation and inflammation were widely involved in both brain and kidney functional processes. Thus, oxidation and inflammation might contribute to CKD-triggered cognitive deficits. Under the condition of continuous adenine treatment, the kidney defense system is disrupted, resulting in the decline of activities of a variety of antioxidant enzymes, such as SOD and GSH-Px. The low molecular weight fucoidan could enhance the activity of antioxidant enzymes and promote the decrease of lipid peroxidation, thus alleviating the symptom of CKD [40]. Nrf2 is released by oxidant stimulation and translocated into the nucleus, interacting with antioxidant response elements and inducing the transcription of downstream SOD, MDA, GSH, and other antioxidant enzyme genes to regulate the expression of antioxidant enzymes. Furthermore, Nrf2 regulated oxidative stress via activating downstream antioxidant proteins, such as HO-1 and NQO1 [41]. Reports have shown that fucoidan attenuated oxidative stress by regulating the gene expression of HO-1 and SOD-1 via the Nrf2 signaling pathway in HaCaT cells [42]. In the current study, we found that fucoidan can improve the activities of SOD and GSH-Px in the kidney and hippocampus and reduce the content of MDA. In addition, our results suggested that fucoidan protected tissue against adenine-induced injury through downregulation of GSK-3β and activation of the Nrf2-HO-1 signaling-pathway-mediated antioxidant responses. Another report evidenced that the possible anti-inflammatory mechanism of fucoidan is the downregulation of MAPK and NF-κB signaling pathways and the following reduction of pro-inflammatory cytokines [43]. Our results indicated that fucoidan significantly inhibited M1 microglia-related mRNA expression of TNF-α, IL-1β, and iNOS, and one the contrary, fucoidan upregulated M2 microglia-related mRNA expression of IL4, CD206, Arg1, and TGFβ. Therefore, we concluded that fucoidan exerts anti-inflammatory effects probably by via inhibiting M1 and promoting M2 microglial polarization.

Reports have shown that the antioxidant and anti-inflammatory activity of fucoidan is dependent on its composition and molecular weight. The higher the content of sulfated polysaccharides and phenolics, the more increased the activity of radical scavenging and antioxidation was observed [44]. In addition, ROS is difficult to be scavenged by fucose, but easy to react with galactose, mannose, and uronic acid, cleave glycosidic bonds, and oxidize uronic acid. Thus, fucoidan removes ROS possibly via reduction reaction [45]. The effect of fucoidan molecular weight on antioxidant activity is controversial. Qi et al. investigated the antioxidant function of fucoidan with different molecular weights (151.7, 64.5, 58.0, and 28.2 kDa) derived from *Ulva pertusa* Kjellm, and found that low molecular weight fucoidan showed significant inhibition on O_2_^−^· and OH [46]. Another study indicated that the relationship between fucoidan molecular weights and their antioxidant activities is not simply linear. The fucoidans with 3.8 kDa, 1.0 kDa, and >8.3 kDa have better hydroxyl radical scavenging activity and antioxidant activity [47]. The anti-inflammatory effects of fucoidan were dramatically increased with a corresponding decrease in molecular mass and increase of the sulfate content [48]. The fucoidan with 5–30 kDa from *Macrocystis pyrifera* showed a highly effective cytokine (TNF-α, IL-1β, and IL-6) inhibition at lower concentrations in LPS-stimulated PBMCs and human THP-1 cells [49]. The antioxidant and anti-inflammatory effects of fucoidan in this study are probably due to the high contents of sulfates and mannose.

In order to explore the hallmarks of memory impairment induced by renal injury, we reanalyzed the genes with large differences in the transcriptome analysis for qPCR analysis, and Pearson correlation with behavior was performed. Camk2n1 and Camk2n2 are endogenous inhibitors of calcium/calmodulin-dependent protein kinase II (CaMKII), a key synaptic signaling molecule for learning and memory [50], which is positively correlated with learning and memory ability. Pro-inflammatory phenotype microglia were polarized via Apod-mediated NLRC4 inflammasome activation, and targeting microglial Apod promoted neural stem cell self-renewal and inhibited neuronal apoptosis [51]. The expression levels of Cd68 and Atp5b were significantly correlated with the neurofibrillary tangle burden in the AD brain and with cognitive ability [52]. Astrocyte-secreted GFAP is regarded as a marker of astrocytosis in neurodegeneration associated with neurological disorders [53,54]. Histological studies showed that activated astrocytes were located in neuroinflammatory amyloid plaques, and a large number of astrocytes were observed near neurofibrillary tangles in autopsy specimens of AD patients [55]. Serpina3 is a serine proteinase inhibitor and an acute phase protein commonly associated with amyloid deposits in AD brains [56]. Osteopontin (SPP1) is a multifunctional matricellular glycoprotein, and it is notably upregulated during the inflammation associated with Alzheimer’s disease and other neurodegenerative conditions [57]. In the Person correlation analysis, the aforementioned genes Apod, Cd68, GFAP, Serpina3, and SPP1 were negatively correlated with cognitive behavior.

The differential genes in kidneys between normal mice and CKD mice were also analyzed by Person correlation analysis with cognition. Cxcl13 is a ligand of CXCR5, and it is significantly secreted under inflammatory conditions [58]. Reports have indicated that the autocrine and paracrine activities of the ANGPTL protein family are associated with angiogenesis and inflammation; one of the members, Angpt7, promotes inflammation by regulating the expression of genes encoding inflammation-associated factors cyclooxygenase-2 (COX-2), inducible nitric oxide synthase (iNOS), tumor necrosis factor alpha (TNF-α), interleukin-1 beta (IL-1β), IL-6, and transforming growth factor beta 1 (TGF-β1) in macrophages [59]. We found that the expressions of Angptl7 and Cxcl13 in the kidney were negatively correlated with cognitive behavior, while the genes expressed in the kidney, such as Slc22a12 and Acy3, are positively correlated with behavior. hURAT1(Slc22a12) is a transporter responsible for renal urate reabsorption; renal hypouricemia will occur in patients with abnormalities in the hURAT1 gene [60]. In kidney proximal tubules, aminoacylase III (Acy3) plays an important role in deacetylating mercapturic acids, and the predominant cytoplasmic localization of Acy3 may explain the greater sensitivity of the proximal straight tubule to the nephrotoxicity of mercapturic acids [61].

## 4. Materials and Methods

### 4.1. Materials

Fucoidan from Laminaria japonica with an average molecular weight of 242 kDa was purchased from Yuanye Biotech Co., Ltd., (Shanghai, China). The carbohydrates of fucoidan were composed of L-fucose (85.4 mol%), D-mannose (2.3 mol%), D-glucuronic acid (3.3 mol%), D-glucose (1.3 mol%), D-xylose (5.8 mol%), and L-Rhamnose (1.9 mol%), as evidenced by high-performance liquid chromatography (HPLC). The content of total sugar was determined as 86.9 ± 2.7%, and 38.9 ± 0.4% of sulfate residues were contained; the detected methods referred to previous report [44]. Adenine (Sigma-A8626) was purchased from Sigma-Aldrich, and losartan potassium tablets were produced from Merck Sharp & Dohme Limited (U.K.). Fucoidan and losartan were dissolved in saline and stored at concentrations of 10, 20, and 2 mg ml^−1^, respectively.

### 4.2. Animals and Experimental Procedure

All animal care and experimental schemes were carried out in accordance with the guidelines of Animal Experiment Committee of Guangdong Ocean University (SYXK2020-0021). ICR mice (male, 34 ± 2 g, 8 weeks old) were purchased from Guangdong Zhiyuan Biomedical Technology Co., Ltd. (Guangzhou, China). The animals were placed in a specific environment (with 12 h of light/dark cycle starting at 7:00 a.m., temperature 22 ± 1 °C, and humidity 55% ± 10%). During the experiment, the animals were fed with powdered SPF grade maintenance rat feed (purchased from Guangdong Medical Experimental Animal Center) and purified water.

The mice were maintained for 7 days before experiment to adapt to the environment, and were randomly divided into 5 groups (*n* = 15 per group), including a control group (saline), vehicle group (saline), low dose of fucoidan group (100 mg kg^−1^ d^−1^, dissolved in saline), high dose of fucoidan group (200 mg kg^−1^ d^−1^, dissolved in saline), and losartan group (Los, positive control, 20 mg kg^−1^ d^−1^, dissolved in saline). Mice in the control group were gavaged with saline and a common feed for 33 days, while mice in vehicle group were oral administered with saline and a diet containing 0.25% adenine. The drug-treated groups were simultaneously gavaged with fucoidan (100 and 200 mg kg^−1^ d^−1^) or losartan (20 mg kg^−1^ d^−1^) and a diet containing 0.25% adenine. The dose of adenine was selected on the basis of previous reports in the literature [62]. To determine the administered doses of fucoidan, we investigated a large number of published literatures, and the dosages ranged from 10 to 500 mg kg^−1^ d^−1^. The dosage of fucoidan in the current study was determined based on a previous report [40]. After a 4-week treatment, the behavioral experiments were performed (continual intragastric administration of the drugs during the behavioral experiment) in the following order: open field experiment, object recognition test, object location test, and passive avoidance test, on day 27, 28, 30, 32, respectively. After the behavioral test, the animals were anesthetized with chloral hydrate, the urine and serum of mice were collected, kidneys (drained and weighed) and hippocampus were taken and stored at −80 °C for subsequent experiments.

### 4.3. Behavioral Test

The open field test is used for evaluating the autonomous behavior, exploratory behavior, and anxiety of experimental animals in novel environments. The experimental method referred to a previous method with minor modification [20]. The object recognition test and object location test were established according to the principle that an animal possesses an innate tendency to explore new objects or novel located objects. The experiments were performed as described previously with minor modification [20]. In the passive avoidance test, the box was divided into a light and dark zone. In the training trial, each mouse was allowed to move freely between the two zones for 1 min and received an electric shock (0.5 mA, 3 s) as soon as it entered the dark zone. The test trial was performed with a 24 h interval, the time of the first electric shock (step-through latency) and the number of electric shocks (number of errors) were recorded within 300 s. The experimental method referred to a previous report with partially modification [63].

### 4.4. Measurement of Urea, Uric Acid, and Creatinine

The blood was naturally coagulated at room temperature for 10–20 min, and the urine was collected with sterile tube, the samples were centrifuged for 20 min (1000 g), and the supernatant was carefully collected. Creatinine (Cr), urea, and uric acid (UA) were quantified using ELISA kit (MEIMIAN, Jiangsu, China) according to manufacturer’s instructions. The absorbances were measured using an ELISA microplate reader (Biotek, Winooski, VT, USA).

### 4.5. RNA-Seq Analysis

The total mRNA was extracted from kidney and hippocampus using the RNA-easy Isolation Reagent following the manufacturer’s protocol. The integrity of RNA samples and DNA contamination and the purity of RNA were determined by agarose gel electrophoresis, NanoPhotometer spectrophotometer, and Agilent 2100 bioanalyzer, respectively, to ensure the sequencing of qualified samples. The gene expression difference was analyzed by software (*p* < 0.05). In order to explore the mechanism of fucoidan in improving memory impairment caused by renal injury, transcriptomic analysis was carried out in brain and kidney tissues of naturally aging model mice (male, 24-month-old) and young mice (male, 3-month-old). DAVID software was used for GO (Gene Ontology) functional annotation and enrichment analysis of significantly different genes obtained by transcriptome analysis.

### 4.6. ELISA Analysis of MDA, SOD, and GSH-Px

After the behavioral experiment, the hippocampus and kidney were made into 10% homogenate with cold saline, and the supernatant after centrifugation was collected to determine the content of MDA, SOD, and GSH-Px activity. The levels of SOD and MDA were evaluated using commercial kits (Nanjing Jiancheng Bioengineering Institute, Nanjing, China) and the GSH-Px was measured by ELISA kits (MEIMIAN, YanCheng, China) according to the manufacturer’s instructions. Finally, the absorbance was measured at 450 nm using a microplate spectrophotometer (Biotek, Winooski, VT, USA).

### 4.7. RNA Isolation and RT-qPCR Analysis

The total mRNA was extracted from kidney and hippocampus using the RNA-easy Isolation Reagent following the manufacturer’s protocol. cDNA was synthesized from 1 µg total RNA using a HiScript II Q Select RT SuperMix for qPCR (+gDNA wiper) kit. PCR amplifications were performed on a CFX96TouchTM Real-Time PCR Detection System using ChamQ Universal SYBR qPCR Master Mix kit at conditions of initial activation at 95 °C for 30 s, followed by 40 cycles of amplification (95 °C for 5 s, 60 °C for 30 s). Gene expression levels were normalized to the RNA expression of housekeeping gene β-Actin. Fold change was calculated using the 2^−∆∆Ct^ method. The amplification primers for genes are summarized in Appendix A.

### 4.8. Statistical Analysis

All data are presented as mean ± SEM. Statistical analysis was performed using GraphPad Prism 9. Statistical comparisons were performed using one- or two-way analysis of variance (ANOVA). *p* < 0.05 was considered as statistically significant. In addition, Pearson correlation analysis was used to analyze the relationship between kidney/hippocampus gene levels and object recognition memory. All correlation statistical analyses were performed by SPSS 22.0.

## 5. Conclusions

The present study first clarified that fucoidan alleviated CKD-triggered memory deficits. GSK3β-Nrf2-HO-1 mediated oxidative signaling and the microglial/macrophage polarization-related inflammatory pathway are likely involved in adenine-induced kidney injury and memory deficits, and are reversed by fucoidan treatment. Additionally, four kidney genes, Acy3, Slc22a12, Angpt7, and Cxcl13, are clarified as hallmarks involved in CKD-triggered memory deficits. This work contributes to reveal the mechanism and new treatment direction of cognitive impairment in patients with CKD and provide an experimental basis for the development of fucoidan as a regulator of CKD-triggered memory deficits.

## Figures and Tables

**Figure 1 marinedrugs-20-00258-f001:**
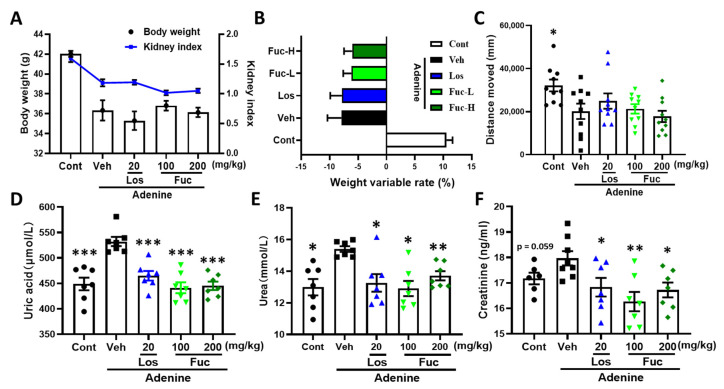
Effects of fucoidan on body and renal indexes in adenine-induced CKD mice. ICR mice (male, 8-week-old) were orally co-administered with fucoidan (Fuc), losartan (Los) or vehicle (saline), and adenine (0.25% contained diets) for 33 days. (**A**) Body weight and kidney index (kidney weight/body weight) among five groups. (**B**) Body weight variable rate (the increased body weight ratio between last and first drug treatment). (**C**) Total distance moved (within 8 min) in the open field test. The expression levels of uric acid (**D**), urea (**E**), and creatinine (**F**) were determined by ELISA kits. The data are expressed as mean ± SEM (*n* = 10). * *p* < 0.05, ** *p* < 0.01, and *** *p* < 0.001 vs. vehicle group (Veh), one-way ANOVA post hoc Dunnett’s test.

**Figure 2 marinedrugs-20-00258-f002:**
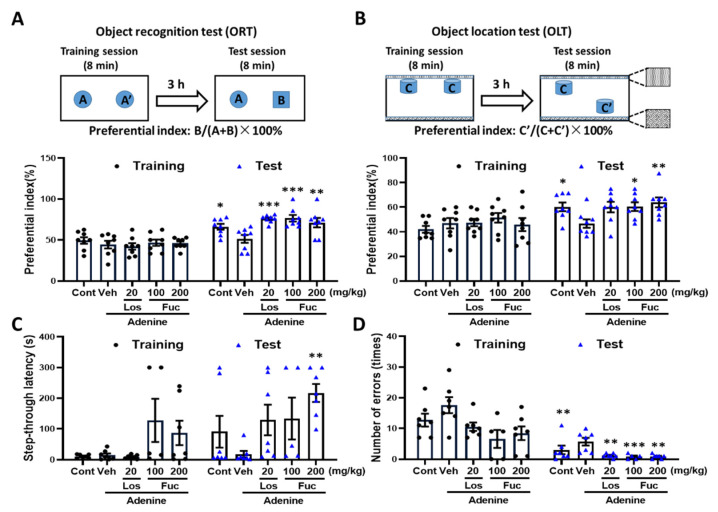
Effects of fucoidan on memory function in adenine-induced CKD mice. (**A**) Object recognition test (ORT). The preferential indexes for the training and test sessions are shown. *p* = 0.017, drug × time interaction was analyzed using repeated measures two-way ANOVA, F(4, 28) = 3.64. * *p* < 0.05, ** *p* < 0.01, and *** *p* < 0.001 vs. vehicle-treated CKD mice, post hoc Dunnett’s test. (**B**) Object location test (OLT). The preferential indexes for the training and test sessions are shown. *p* = 0.0998, drug × time interaction was analyzed using repeated measures two-way ANOVA, F(4, 56) = 2.05. * *p* < 0.05, ** *p* < 0.01 vs. vehicle-treated CKD mice, post hoc Dunnett’s test. (**C**) The step-through latency entering the darkroom in passive avoidance memory test. *p* = 0.288, drug × time interaction was analyzed using repeated measures two-way ANOVA, F(4, 56) = 1.28. ** *p* < 0.01 vs. vehicle-treated CKD mice, post hoc Dunnett’s test. (**D**) The number of errors into the darkroom were measured. *p* = 0.482, drug × time interaction was analyzed using repeated measures two-way ANOVA, F(4, 44) = 0.88. ** *p* < 0.01, *** *p* < 0.001 vs. vehicle-treated CKD mice, post hoc Dunnett’s test. The data are expressed as mean ± SEM (*n* = 8).

**Figure 3 marinedrugs-20-00258-f003:**
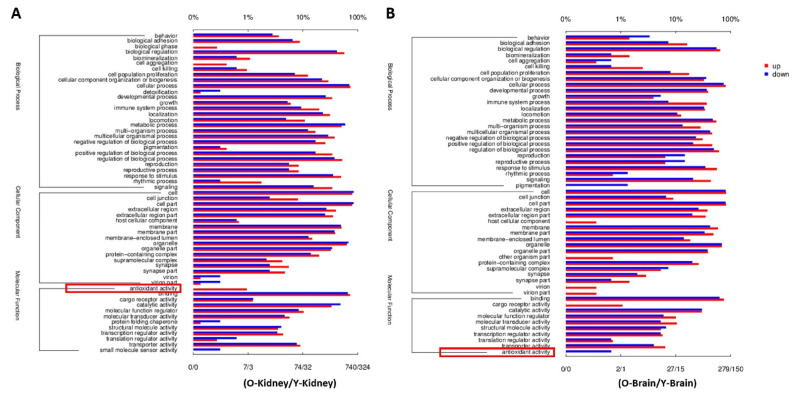
Column chart of GO annotation in the brain and kidney of CKD mice. GO analysis in the kidney (**A**) and brain (**B**) between aged-CKD and young mice was performed. The data are expressed as mean ± SEM (*n* = 3).

**Figure 4 marinedrugs-20-00258-f004:**
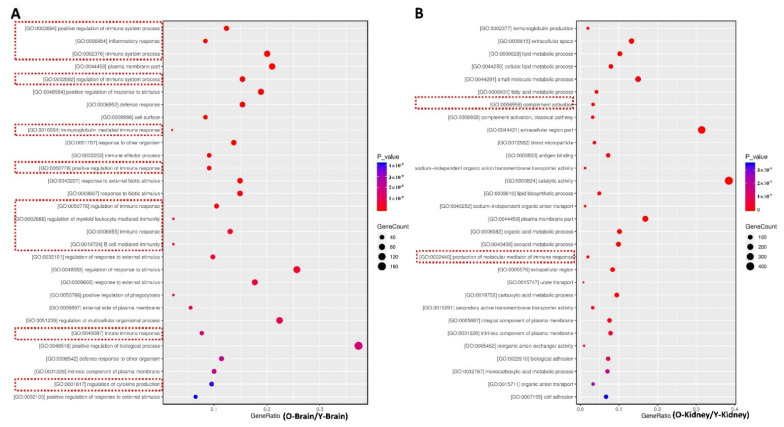
Bubble diagram of GO enrichment analysis of differential genes. GO enrichment analysis in the brain (**A**) and kidney (**B**) between aged-CKD and young mice was performed. The data are expressed as mean ± SEM (*n* = 3).

**Figure 5 marinedrugs-20-00258-f005:**
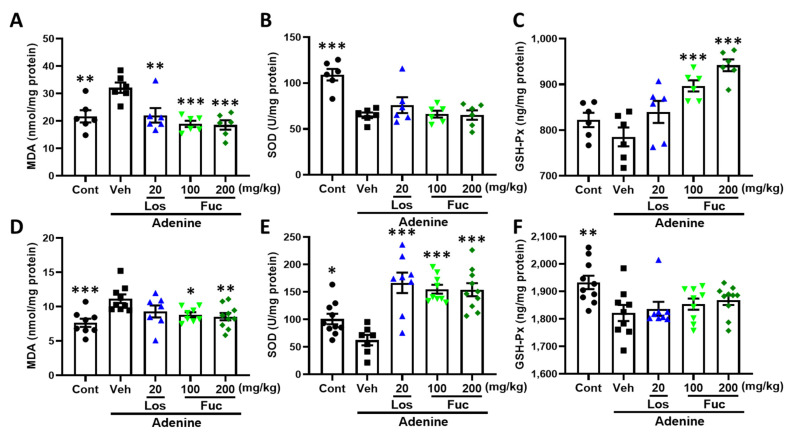
Effects of fucoidan on oxidative indexes in adenine-induced CKD mice. (**A**–**C**) The expression level of MDA and activities of SOD and GSH-Px were measured in mice hippocampus. (**D**–**F**) The expression level of MDA and activities of SOD and GSH-Px were measured in mice kidney. The data are expressed as mean ± SEM (*n* = 6–10). * *p* < 0.05, ** *p* < 0.01, and *** *p* < 0.001 vs. vehicle group (Veh), one-way ANOVA post hoc Dunnett’s test.

**Figure 6 marinedrugs-20-00258-f006:**
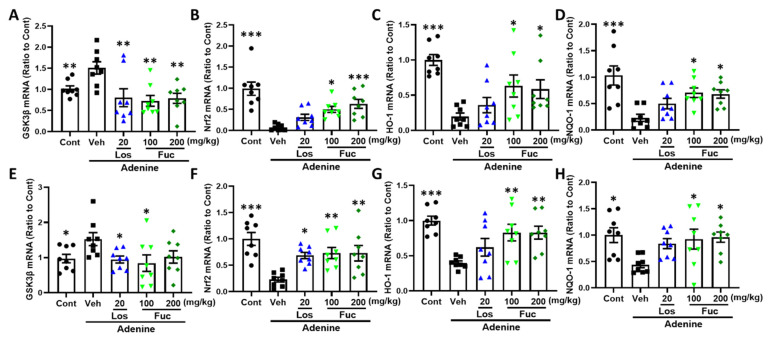
Fucoidan regulated GSK3β-Nrf2-HO-1 signaling pathway in hippocampus and kidney of adenine-induced CKD mice (*n* = 8). (**A**–**D**) The mRNA expression of GSK3β, Nrf2, HO-1, and NQO-1 in the hippocampus. (**E**–**H**) The mRNA expression of GSK3β, Nrf2, HO-1, and NQO-1 in the kidney. * *p* < 0.05, ** *p* < 0.01, and *** *p* < 0.001 vs. vehicle group (Veh), one-way ANOVA post hoc Dunnett’s test.

**Figure 7 marinedrugs-20-00258-f007:**
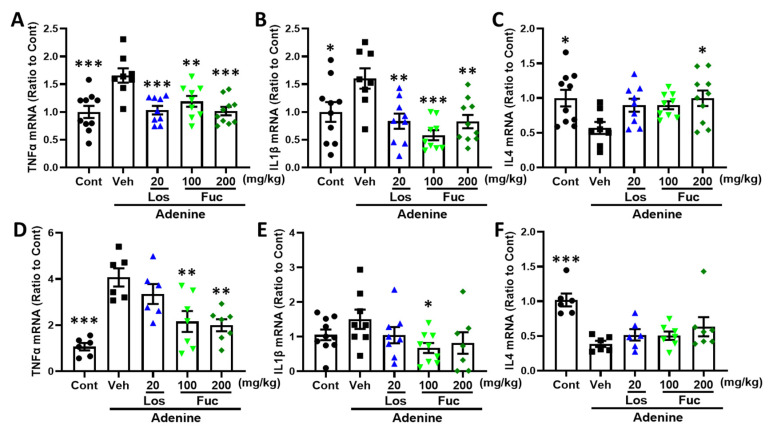
Effects of fucoidan on inflammatory factors in hippocampus and kidney of adenine-induced CKD mice (*n* = 6–10). (**A**–**C**) TNFα, IL-1β, and IL-4 mRNA expression in the hippocampus; (**D**–**F**) TNFα, IL-1β, and IL-4 mRNA expression in the kidney. * *p* < 0.05, ** *p* < 0.01, and *** *p* < 0.001 vs. vehicle group (Veh), one-way ANOVA post hoc Dunnett’s test.

**Figure 8 marinedrugs-20-00258-f008:**
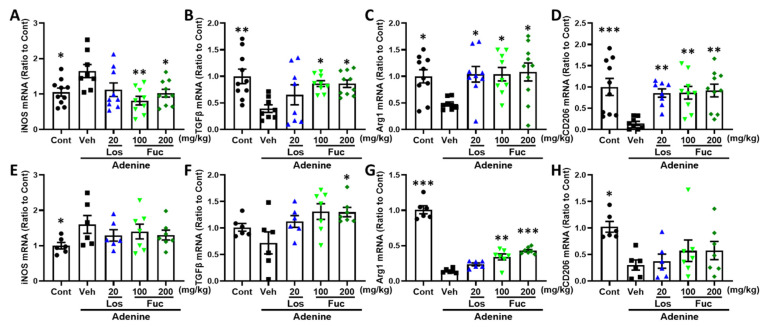
Effects of fucoidan on M1/M2 microglia or macrophage polarization in hippocampus and kidney of adenine-induced CKD mice (*n* = 6–10). (**A**–**D**) iNOS, TGFβ, Arg1, and CD206 mRNA expression in the hippocampus; (**E**–**H**) iNOS, TGFβ, Arg1, and CD206 mRNA expression in the kidney. * *p* < 0.05, ** *p* < 0.01, and *** *p* < 0.001 vs. vehicle group (Veh), one-way ANOVA post hoc Dunnett’s test.

**Figure 9 marinedrugs-20-00258-f009:**
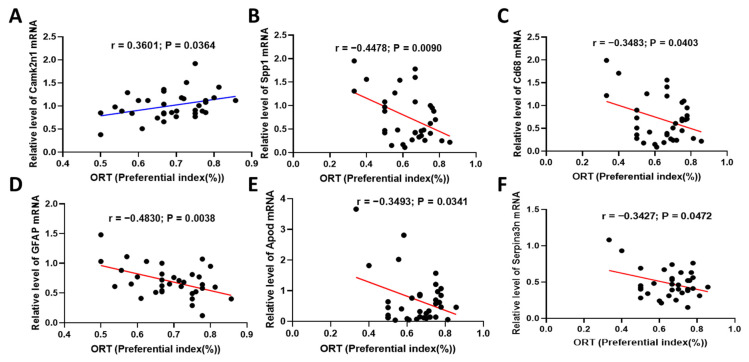
Pearson correlation analysis between hippocampus gene expression and cognitive behavior. The mRNA qPCR analysis of Camk2n1 (**A**), SPP1 (**B**), Cd68 (**C**), GFAP (**D**), Apod (**E**), and Serpina3 (**F**) were performed.

**Figure 10 marinedrugs-20-00258-f010:**
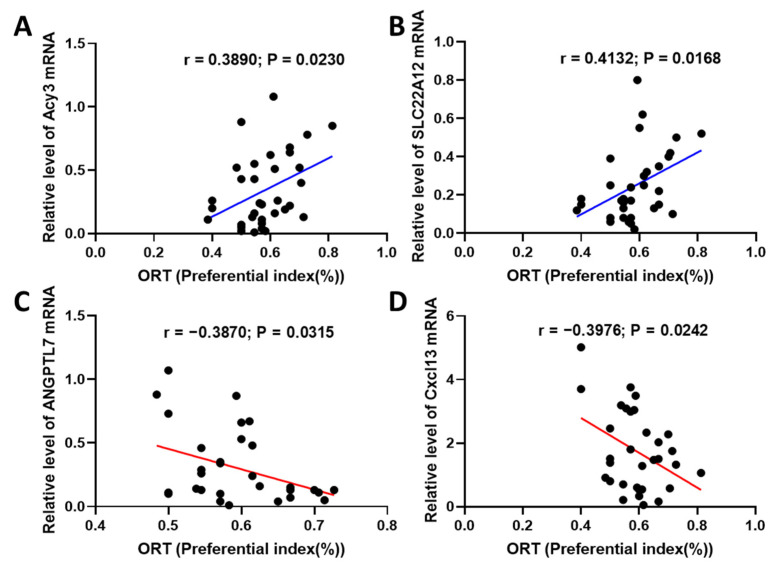
Pearson correlation analysis between kidney gene expression and cognitive behavior. The mRNA qPCR analysis of Acy3 (**A**), Slc22a12 (**B**), Angptl7 (**C**), and Cxcl13 (**D**) were performed.

## Data Availability

Not applicable.

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
