# Peer review of "The Emerging Evidence for a Protective Role of Fucoidan from Laminaria japonica in Chronic Kidney Disease-Triggered Cognitive Dysfunction"

_marinedrugs, 2022, doi:10.3390/md20040258_

Round 1

Reviewer 1 Report

Dear authors! I have read the sent manuscript and I have some questions and recommendations.

  1. In section 4, please indicate the source of fucoidan, its monosaccharide composition, degree of sulfation, molecular weight, polyphenol content, reproductive phase of the algae. It is known that the composition of fucoidan affects its properties (https://doi.org/10.3390/md18050275). Please give the title of the manuscript in accordance with the data.
  2. In section 4.1. indicate the solvent for the introduction of fucoidan. Based on what data doses for administration to animals are selected. Give justification.
  3. For sections 4.3 and 4.5, indicate the brand, manufacturer of the analysis device. 
  4. Inflammation is involved in various human diseases. Anti-inflammatory mechanisms described for fucoidan include free radical scavenging (eg, https://doi.org/10.3390/md18050275), suppression of nitric oxide production, tumor necrosis factor-alpha (TNF-α), prostaglandin E2, interleukin-1 beta and interleukin-6 (for example, https://doi.org/10.3390/md19050277) and others. Discuss the effect of composition, molecular weight on the activity of fucoidan.
  5. Improve on conclusion.

Author Response

Response to Reviewer 1 Comments

Point 1: In section 4, please indicate the source of fucoidan, its monosaccharide composition, degree of sulfation, molecular weight, polyphenol content, reproductive phase of the algae. It is known that the composition of fucoidan affects its properties (https://doi.org/10.3390/md18050275). Please give the title of the manuscript in accordance with the data.

Response 1: Thanks for your valuable suggestions, we determined the compositions of fucoidan as the way you recommended in (https://doi.org/10.3390/md18050275), and referred to this paper in L383-389. The detailed information of materials was also described in section 4.1 and marked in red color.

Point 2: In section 4.1. indicate the solvent for the introduction of fucoidan. Based on what data doses for administration to animals are selected. Give justification.

Response 2: Thanks for your comment. The fucoidan was dissolved in saline and was intragastric to mice. For the dose selection, we investigated a large number of published literatures, and the dosage is ranged from 10 to 500 mg/kg. The dosage of fucoidan in our study was determined based on previous report (L408-412). In addition, we investigated the low dose (10 mg/kg) of fucoidan on adenine-triggered memory deficits, and found that the memory function was also significantly improved (data not shown) (L123-125).

Point 3: For sections 4.3 and 4.5, indicate the brand, manufacturer of the analysis device. 

Response 3: The information of the brand, and manufacturer of the analysis device were added in the manuscript marked with red color (L438-439, L458-459).

Point 4: Inflammation is involved in various human diseases. Anti-inflammatory mechanisms described for fucoidan include free radical scavenging (eg, https://doi.org/10.3390/md18050275), suppression of nitric oxide production, tumor necrosis factor-alpha (TNF-α), prostaglandin E2, interleukin-1 beta and interleukin-6 (for example, https://doi.org/10.3390/md19050277) and others. Discuss the effect of composition, molecular weight on the activity of fucoidan.

Response 4: Thank you for your excellent comments. We have discussed the composition and molecular weight on the activity of fucoidan in L326-344.

Point 5: Improve on conclusion.

Response 5: Thank you for your comments. We have improved the conclusion in L478-483.

Reviewer 2 Report

  1. Information about control drugs is missing. Authors should add a separate section in material and methods about the drugs.
  2. From where do authors get fucoidan? What was the purity level? Please include the information in the material and methods.
  3. How did the authors determine the treatment doses of the control drug and fucoidan?
  4. Authors should check the fucoidan effect using different doses. Please justify/discuss.
  5. What is the difference between control and vehicle?
  6. There is a huge difference between dosed fucoidan (100 and 200 mg/kg) and losartan (20 mg/kg). Fucoidan doses are too high to be considered for therapeutic purposes. Please justify.
  7. What about cytotoxicity of fucoidan? There is no experiment to check the cytotoxicity of fucoidan. This is a must.
  8. Authors should perform a survival experiment with high fucoidan doses (100 and 200 mg/kg) to confirm its cytotoxicity.
  9. Why author used oral gavage as the route of drug administration? The author needs to experiment using other routes of drug administration to check the efficacy of fucoidan.

Author Response

Response to Reviewer 2 Comments

Point 1: Information about control drugs is missing. Authors should add a separate section in material and methods about the drugs

Response 1: Thanks for your comment, the detailed information of materials was added in section 4.1 and marked in red color (L389-392).

Point 2: From where do authors get fucoidan? What was the purity level? Please include the information in the material and methods.

Response 2: Thanks for your valuable suggestions, we have determined the compositions and purity of fucoidan in section 4.1, and the detailed information was described in L383-389.

Point 3: How did the authors determine the treatment doses of the control drug and fucoidan?

Response 3: For the dose selection, we investigated a large number of published literatures, and the dosage is ranged from 10 to 500 mg/kg. The dosage of fucoidan in our study was determined based on previous report (L408-412). In addition, we investigated the low dose (10 mg/kg) of fucoidan on adenine-triggered memory deficits, and found that the memory function was also significantly improved (data not shown) (L123-125).

Point 4: Authors should check the fucoidan effect using different doses. Please justify/discuss.

Response 4: In current study, we investigated 100 and 200 mg/kg doses of fucoidan according to the published literatures. In addition, we investigated the low dose (10 mg/kg) of fucoidan on adenine-triggered memory deficits, and found that the memory function was also significantly improved (data not shown) (L123-125).

Point 5: What is the difference between control and vehicle?

Response 5: The mice were randomly divided into 5 groups: control group (Cont), vehicle group (Veh), low-dose fucoidan group (100 mg kg-1 d-1), high-dose fucoidan group (100 mg kg-1 d-1), and losartan group (Los, positive control). Except for the normal control group, the other 4 groups of mice were simultaneously given a diet contain 0.25% adenine. The information was added in L403-407. In addition, we reorganized the figures to distinguish different groups.

Point 6: There is a huge difference between dosed fucoidan (100 and 200 mg/kg) and losartan (20 mg/kg). Fucoidan doses are too high to be considered for therapeutic purposes. Please justify.

Response 6: According to a large number of literatures investigated, the administration of fucoidan is mainly oral or intragastric, and the dosage is range from 10 to 500 mg/kg. The dosage of fucoidan in our study was determined as the middle dose of reports (https://doi.org/10.1016/j.carbpol.2010.11.055). The dosage of losartan in our study was determined based on references (eg, https://doi.org/10.1016/j.mvr.2020.104037 ; https://doi.org/10.1016/j.jep.2021.114189 ). In addition, we explored the low dose (10 mg/kg) of fucoidan on adenine-triggered memory deficits in another study, and found that the memory function was also significantly improved (data not shown) (L123-125).

Point 7: What about cytotoxicity of fucoidan? There is no experiment to check the cytotoxicity of fucoidan. This is a must.

Response 7: Thank you for your good advice, we previously carried out experiments to evaluate the cytotoxicity of fucoidan on neuronal cells. Compared with the control group, there was no significant change in the survival rate of SH-SY5Y cells in fucoidan group range from 0.1 to 1000 μg/mL (L85-86). With the increase of fucoidan concentration, 5-30 mg/mL fucoidan treatment significantly reduced the SH-SY5Y cell viability, indicating that high concentration of fucoidan has certain toxicity to cells (https://doi.org/10.3969/j.issn.1673-9159.2021.04.009).

Point 8: Authors should perform a survival experiment with high fucoidan doses (100 and 200 mg/kg) to confirm its cytotoxicity.

Response 8: In current study, we recorded locomotor function after 27 days of fucoidan treatment. No significant locomotor differences were detected in the distances moved between fucoidan and vehicle treated groups (Figure 1C). The mice body weights were also recorded during the drug administration, with no significant changes observed in fucoidan treated groups compared with vehicle group (Supplementary Figure 1). The results indicated that fucoidan treatment (100 and 200 mg/kg) did not significantly affect mice general healthy and behavior (L108-112).

Point 9: Why author used oral gavage as the route of drug administration? The author needs to experiment using other routes of drug administration to check the efficacy of fucoidan.

Response 9: Thank you for your excellent suggestion, although intraperitoneal injection, icv injection, and iv injection are available drug administration routes, oral gavage is a usual way to test the effects of molecules, especially for polysaccharides. A variety of studies indicated that polysaccharides regulated intestine microbiota after oral administration and following interfere some mental diseases via brain-microbiota axis, such as the clinical approved drug, GV971, for Alzheimer’s disease treatment. However, it is important to use other administration way to confirm the efficacy of fucoidan, and we will investigate it in the future study.

Reviewer 3 Report

In the study titled “The Emerging Evidence for A Protective Role of Fucoidan in Chronic Kidney Disease Triggered Cognitive Dysfunction” the authors investigated the ability for Fucoidan to limit neurological alterations in an animal model of chronic kidney disease. Overall the author’s results suggest that Fucoidan treatment limited the progression of CKD-induced memory deficits through the limitation of inflammatory signaling.

Review comments

  1. It would be helpful to the readers to include in the results section 2.1 and/or figure 1 (potentially a line in the to denote adenosine treatment) to specify that the control group were not treated with adenosine. The current description does not indicate this.
  2. Sections 2.2, 2.3, 2.4, 2.5, and 2.6 have the same title: “Fucoidan Ameliorated Cognitive Deficits in Adenine-Induced CKD Mice”. These need to be altered to describe each section appropriately.
  3. The authors noted in section 2.3 that they used 24-month old mice to mimic CKD conditions for RNA-seq analysis. Were any additional analysis performed to confirm that these mice had any CKD related issues prior to being used as a CKD-group? (based upon the wording in this section, I am assuming that these animals were not treated with adenosine) If not how does this represent a true CKD group?
  4. Based upon the authors data, Fucoidan and Losartan treatment groups do not appear to be significantly different, suggesting that treatment with Losartan would also be a potential therapeutic agent to limit CKD mediated memory deficits. This is not addressed in the article and why it might be beneficial to use Fucoidan vs Losartan.

Author Response

Response to Reviewer 3 Comments

Point 1: It would be helpful to the readers to include in the results section 2.1 and/or figure 1 (potentially a line in the to denote adenosine treatment) to specify that the control group were not treated with adenosine. The current description does not indicate this.

Response 1: Thank you for your suggestion, we have reorganized the all figures, and adenine treatment was marked to distinguish different groups.

Point 2: Sections 2.2, 2.3, 2.4, 2.5, and 2.6 have the same title: “Fucoidan Ameliorated Cognitive Deficits in Adenine-Induced CKD Mice”. These need to be altered to describe each section appropriately.

Response 2: We are sorry for the mistake that we shouldn’t make, the subtitles have been revised according to the corresponding contents.

Point 3: The authors noted in section 2.3 that they used 24-month old mice to mimic CKD conditions for RNA-seq analysis. Were any additional analysis performed to confirm that these mice had any CKD related issues prior to being used as a CKD-group? (based upon the wording in this section, I am assuming that these animals were not treated with adenosine) If not how does this represent a true CKD group?

Response 3: Thank you for your valuable question. We evaluated the CKD characteristics before RNA-seq analysis, the urine was collected and ELISA analysis was performed, compared with young mice, the selected 3 aging mice showed significant increased expression of urine acid and urea (Supplementary Figure 2), and the spatial memory function was also decreased in the Morris Water Maze test (data not shown).

Point 4: Based upon the authors data, Fucoidan and Losartan treatment groups do not appear to be significantly different, suggesting that treatment with Losartan would also be a potential therapeutic agent to limit CKD mediated memory deficits. This is not addressed in the article and why it might be beneficial to use Fucoidan vs Losartan.

Response 4: Thank you for your question, it is a good point. Commonly reported side effects of losartan including asthenia, chest pain, diarrhea, fatigue, and hypoglycemia. While, human clinical studies using orally ingested fucoidan have shown no toxicity at a dose of 1 g per day up to three months, or 3 g per day for 12 days (Exp. Hematol. 2007; 35:989–994.), which indicates the safe advantage of fucoidan than losartan. In addition, in another parallel study, we explored the low dose (10 mg/kg) of fucoidan on adenine-triggered memory deficits, and found that the memory function was also significantly improved (data not shown) (L123-125). The results of HE and Masson staining on renal tissues indicated that fucoidan treated group showed more decreased infiltrated inflammatory cells than losartan group (data not shown).

Round 2

Reviewer 1 Report

The authors took into account my recommendations and made the necessary corrections. I have no more questions.

Author Response

Thank you so much for your sincere suggestions. We have revised the language thoroughly according to the advice of a native speaker.

Reviewer 2 Report

What are the control and vehicle groups? Authors need to state clearly. 

Author Response

Response to Reviewer 2 Comments

Point 1: What are the control and vehicle groups? Authors need to state clearly.

Response 1: Thanks for your comment, mice in the control group were gavaged with saline and a common feed for 33 days, while mice in vehicle group were oral administered with saline and a diet contain 0.25% adenine. The drug-treated groups were simultaneously gavaged with fucoidan (100 and 200 mg kg-1 d-1) or losartan (20 mg kg-1 d-1) and a diet contain 0.25% adenine (L439-443).

Reviewer 3 Report

Reviewer Response #4: Based upon the authors response, they are suggesting that Fucoidan does not result in any adverse effects and they are only indicating this from a publication with a total of 37 human subjects (Fucoidan was only administered for up to three months). How is this enough data to suggest that this sulfated polysaccharide does not have adverse effects when compared to a currently approved/used drug?

Additionally the current incidence of the adverse effects the authors have listed (a copy and paste from a google search) for losartan is generally limited to <2% (without additional complications, such as type 2 diabetes); therefore, to this reviewer, the authors current rationale does not apply. 

At this time, the data presented in the current study does not indicate a clear advantage for the suggestion that Fucoidan has increased benefits over losartan. This should be stated in the discussion as more information is still needed to clearly suggest what the authors are suggesting. 

Author Response

Response to Reviewer 3 Comments

Point 1: Reviewer Response #4: Based upon the authors response, they are suggesting that Fucoidan does not result in any adverse effects and they are only indicating this from a publication with a total of 37 human subjects (Fucoidan was only administered for up to three months). How is this enough data to suggest that this sulfated polysaccharide does not have adverse effects when compared to a currently approved/used drug?

Additionally, the current incidence of the adverse effects the authors have listed (a copy and paste from a google search) for losartan is generally limited to <2% (without additional complications, such as type 2 diabetes); therefore, to this reviewer, the authors current rationale does not apply. 

At this time, the data presented in the current study does not indicate a clear advantage for the suggestion that Fucoidan has increased benefits over losartan. This should be stated in the discussion as more information is still needed to clearly suggest what the authors are suggesting. 

Response 1: Thank you for your suggestion, in current study, a relatively high concentration of fucoidan was adopted, thus, the toxicity become an issue that cannot be ignored. An acute toxic study of fucoidan from Sargassum Wightii grevillea in wistar rats showed that 2000 mg kg-1 dose of fucoidan did not induce any remarkable toxic signs or mortality. While, in the subacute toxicity study, except for a reduction of serum glucose and cholesterol, no significant changes of biochemical, haematological parameters, or toxic signs at organ levels were observed after daily oral administration of 100-400 mg kg-1 fucoidan for 28 days [Ramu, et al., Toxicol. Rep. 2020]. The 28 days oral administration of low molecular weight of fucoidan (<667 Da) from Laminaria japonica up to 2000 mg kg-1 caused no toxicological indications in Sprague-Dawley rats [Hwang et al., Mar. Drugs 2016]. However, the 6 months treatment of fucoidan from Laminaria japonica at doses of 900 and 2500 mg kg-1 significantly prolonged the clotting time [Li et al., Food Chem. Toxicol. 2005], the same phenomenon was observed after gavage of fucoidan from Cladosiphon okamuranus over 1200 mg kg-1 for 3 months [Gideon et al., J. Med. Food 2008]. The aforementioned differential results indicate that the potential toxicity of fucoidan depends on the originated sources, doses, as well as the molecular weights. The safety of fucoidan was also evaluated in humans, no changes in the liver and kidneys parameters was observed after oral taken of 4.05 g per day of mozuku fucoidan derived from Cladosiphon okamuranus [Citkowska et al., Mar. Drugs 2019]. Our results indicated that 33 days oral administration of fucoidan (200 mg kg-1) did not affect the body weights and locomotor activity compared with adenine-treated vehicle group. Nevertheless, the acute and subacute toxicity should be investigated in the future study. Losartan, an angiotensin II type 1 (AT1) receptor antagonist, was approved by FDA in 1995 for hypertension and renal disease. Recently, losartan has been demonstrated exerting neurogenesis, astrocytes motility, and memory improvement effects in APP/PS1 AD model mice [Drews et al., Pharmaceuticals (Basel) 2021]. While, losartan significantly improved memory function in aged 75–89 years elderly hypertensive patients [Fogari et al., J. Hum. Hypertens. 2003]. Intriguingly, 30 days treatment of losartan to normal rat led to dendritic spine loss, tau hyperphosphorylation, and memory deficits [Liu et al., J. Alzheimers Dis. 2014]. The findings indicate that losartan is a double-edge sword to memory function. Our study demonstrated that losartan (20 mg kg-1) treatment reversed adenine-induced memory deficits. Although low dose of fucoidan (10 mg kg-1) in our parallel study showed memory attenuation in adenine-induced CKD mice (data not shown), a further investigation should be performed to compare the effect of fucoidan and losartan on cognitive function. We added these discussions in L300-330.

Round 3

Reviewer 3 Report

The authors need to change line 324-325 to state that the 30 day treatment of Losartan was performed to induce a state of hypotension in these animals, as this study was performed to investigate the relationship between dementia and low blood pressure. These were no longer "normal rats" since they induced hypotension! This would suggest an important relationship in controlling BP when using losartan. 

Author Response

Reply to Reviewer 3
Q 1: The authors need to change line 324-325 to state that the 30 day treatment of Losartan was performed to induce a state of hypotension in these animals, as this study was performed to investigate the relationship between dementia and low blood pressure. These were no longer "normal rats" since they induced hypotension! This would suggest an important relationship in controlling BP when using losartan.

Answer: Thank you for your constructive comment, we revised the sentence as “30 days treatment of losartan led to a hypotension model in rats, and promoted dendritic spine loss, tau hyperphosphorylation, and memory deficits, indicating blood pressure as a considerable factor when taking losartan” (L323-326).